# Exploring the Formation of Polymers with Anti-Amyloid Properties within the 2′3′-Dihydroxyflavone Autoxidation Process

**DOI:** 10.3390/antiox11091711

**Published:** 2022-08-30

**Authors:** Andrius Sakalauskas, Agne Janoniene, Gediminas Zvinys, Kamile Mikalauskaite, Mantas Ziaunys, Vytautas Smirnovas

**Affiliations:** Institute of Biotechnology, Life Sciences Center, Vilnius University, LT-10257 Vilnius, Lithuania

**Keywords:** aggregation, amyloid-beta, alpha-synuclein, flavone, inhibition, autoxidation, polymers

## Abstract

Amyloid-β and α-synuclein aggregation into amyloid fibrils is linked to the onset and progression of Alzheimer’s and Parkinson’s diseases. While there are only a few disease-modifying drugs, it is essential to search for new, more effective ways to encounter these neurodegenerative diseases. Multiple research articles have shown that the autoxidation of flavone is a critical factor for activating the inhibitory potential against the protein aggregation. Despite this, the structure of the newly-formed inhibitors is unknown. In this research, we examined the autoxidation products of 2′,3′-dihydroxyflavone that were previously shown to possess one of the most prominent inhibitory effects against amyloid-β aggregation. Their analysis using HPLC suggested the formation of polymeric molecules that were isolated using a 3 kDa cut-off. These polymeric structures were indicated as the most potent inhibitors based on protein aggregation kinetics and AFM studies. This revelation was confirmed using MALDI-TOF and NMR. We also show that active molecules have a tendency to reduce the Amyloid-β and α-synuclein aggregates toxicity to SH-SY5Y cells.

## 1. Introduction

Protein accumulation into insoluble aggregates is linked with neurological disorders, such as Alzheimer’s disease (AD) and Parkinson’s disease (PD) [1]. AD was recognized as the most prevalent neurological condition affecting over 50 million people worldwide [2] and is projected to increase to 76 million by 2030 [3]. The pathological hallmark of AD is the deposition of extracellular amyloid-β (Aβ) peptide plaques and Tau neurofibrillary tangles in neuronal cells [4]. PD is known to be the second most common neurodegenerative disease after AD [5]. PD is considered a movement disorder caused by the accumulation of neuronal inclusions (Lewy Bodies) consisting of α-synuclein (aSyn) aggregates [6,7]. While both disorders are associated with aggregation that causes the onset of the disease, it is essential and beneficial to search for a potential treatment that could affect the beginning and progression of AD and PD.

Many possible counters for these neurological disorders involve treatment using anti-amyloid drugs [8,9,10,11]. In order to be effective, these compounds should selectively bind to the aggregation-prone peptide or protein, stabilizing them or changing the aggregation pathway to form non-toxic amorphous aggregates [12]. With this knowledge, numerous different compound groups have been screened, sorting out only potential inhibitors. However, this number has been vastly reduced in the clinical trials leaving only a few that can be used in the symptomatic treatment against AD and PD [13,14]. Therefore, it is necessary to fully test and categorize the newly found inhibiting molecules and understand their practical sense as anti-amyloid compounds.

Flavones belong to a group of natural anti-oxidants that are found in nature in fruits, herbs, spices, and vegetables [15,16]. Besides their anti-microbial, anti-HIV, anti-cancer, anti-platelet, neuroprotective, anti-mutagenic, anti-allergic and anti-inflammatory characteristics [17,18,19], they were also shown in multiple reports to possess anti-amyloid properties [20,21,22]. A large number of flavone derivatives are described as acetylcholinesterase (AChE) inhibitors which makes them appealing candidates for the symptomatic treatment of AD [23,24]. In fact, there are twice as many flavone AChE inhibitors than flavanones, meaning that the structural integrity of the C2-C3 bond is essential [25]. However, the stability of flavones is questionable. The derivatives possessing neighboring hydroxyl groups are very likely to autoxidize at neutral or basic pH, completely changing the nature of these molecules [21]. Nevertheless, autoxidation improves the inhibitory effect against Aβ and insulin aggregation [20,22]. This effect is shown in studies covering different polyphenolic compound groups including flavones. The most famous is EGCG that possesses an inhibitory effect against aggregation of different proteins [26,27]. While the compound autoxidation mixture seems to have anti-amyloid features, it can hold components that have opposite or even cytotoxic effects. Therefore, it is essential to understand which autoxidation products exhibit inhibitory features towards protein aggregation.

The autoxidation of polyphenolic molecules can lead to multiple distinct products obtaining unique structures and features [28,29]. This process involves degradation, oxidative coupling [17,30,31], and polymerization [32]. A recent study shows that the browning of the epicatechin and epigallocatechin samples is related to the oxidative coupling where o-quinone intermediates participate [33]. A similar pattern is observed in the iron-mediated autoxidation of flavonoids that leads to the formation of dehydro-type dimers, while the autoxidative degradation results in the breakdown of the C ring and the formation of HBA, DHBA, THBA, and THPGA derivatives [34].

In order to identify the inhibitory molecules in the flavone oxidation mixture, we selected oxidized 2′,3′-dihydroxyflavone, which has previously been shown to possess the highest inhibitory potential among number of tested flavones against Aβ aggregation [21]. The 2′,3′-dihydroxyflavone has a C2-C3 double bond catechol moiety and the absence of 3-hydroxyl group that should favor the stabilization of the o-quinone and the oxidation coupling [30]. In this work, we show that the 2′,3′-dihydroxyflavone oxidation products range from small degradation to large oligomeric molecular compounds. The sample molecules were separated using a 3 kDa cut-off and examined using HPLC. The anti-aggregation effects were tested on Aβ and aSyn aggregation. The fraction containing inhibitor molecules was scanned using MALDI-TOF and NMR.

## 2. Materials and Methods

**2′,3′-dihydroxyflavone oxidation.** The 2′3′-dihydroxyflavone stock solution was prepared by dissolving 2′,3′-dihydroxyflavone (Indofine Chemical Company, Inc., Hillsborough Township, NJ, USA) in dimethylsulfoxide (DMSO, Carl Roth) to a final concentration of 10 mM. The oxidation solution was prepared by diluting 10 mM stock solution with 10 mM sodium phosphate buffer (pH 8.0) and DMSO to yield a final flavone concentration of 0.5 mM in 9 mM sodium phosphate buffer solution containing 10% DMSO. The final 10% DMSO in buffer solution was used to increase the solubility. The autoxidation was carried out by incubating the solution in 37 °C for 100 h.

**Separation of oDHF fractions.** The oxidized 2′3′-dihydroxyflavone (oDHF) was distributed to the concentrators with a 3 kDa cut-off. The concentrators were spun at 4000 g for 20 min. The flowthrough consisting of <3 kDa molecular weight molecules (oDHF_LW_) and a cut-off sample consisting of >3 kDa molecular weight molecules (oDHF_HW_) were separated. The oDHF_HW_ fraction was washed using the 9 mM sodium phosphate buffer solution containing 10% DMSO (pH 8.0) (further referred to as oxidation buffer) to remove all low molecular weight compounds from the mixture. The oDHF_HW_ was diluted to the original volume of oDHF to retain a comparable concentration of high molecular weight molecules in the sample (Appendix A displays the sample preparation procedure).

**High-performance liquid chromatography.** The oDHF and its fractions (oDHF_LW_ and oDHF_HW_) were separated and analyzed using Shimadzu UFLC system with a CMB-20A communication module, two LC20AD quaternary and isocratic pumps, a SIL-20AC autosampler, a CTO-20A column compartment and an SPD-M20A DAD detector (Shimadzu Corp., Kyoto, Japan). For the detection of the eluting molecules, the DAD spectra recording was set from 190 nm to 500 nm with a data rate of 12.5 Hz. The ODS-AQ HPLC separation column (15 × 4.6, 3 µm, YMC) was used together with a 1 cm guard column. The HPLC grade MeCN (Fisher Scientific) and Milli-Q water (18.2 MΩ cm^−1^, Milli-Q Plus system, Millipore Bedford, MA, USA) were used for the RP-HPLC separation.

The samples were separated using a trinary gradient consisting of ultrapure water (eluent A), MeCN (eluent B) and 1% TFA in ultrapure water (eluent C). A constant 10% flow of eluent C was used to maintain 0.1% TFA concentration in the column throughout the separation experiment. The gradient between eluents A and B was 18% (0 min), 81% (20 min), and 81% (28 min). Before each analytical run, the column equilibration (10 column volume) was performed. The column thermostat was set to 40 °C and the flow rate to 1 mL min^−1^.

**Solid state 1H NMR spectroscopy.** 1H solid-state NMR spectra were recorded on a Bruker spectrometer (400 MHz) (Billerica, MA, USA) in DMSO-D6 using residual DMSO signals (2.50 ppm) as the internal standard, and proton chemical shifts were expressed in parts per million (ppm).

**MALDI-TOF mass analysis.** Full scan MALDI-TOF-MS mass spectra were performed on a Bruker Autoflex Speed (Billerica, MA, USA) using a 1 kHz smartbeam™-II laser for MALDI. The mass spectra laser shots were collected for the spectra at a 19.5 kV acceleration voltage. The oDHF_HW_ sample was prepared with 2,5-dihydroxybenzoic acid (2,5-DHB) and sinapinic acid (SA) matrixes.

Preparation of sample with 2,5-DHB matrix: 1 µL of 2,5-DHB solution (20 mg/mL in the 30% ACN [*v*/*v*] containing 0.1% TFA) and 1 µM of oDHF_HW_ were deposited on the MALDI target plate and dried under vacuum.

Preparation of sample with SA matrix: saturated SA in EtOH was deposited on the MALDI target plate and dried under vacuum. Then 1 µL of SA solution (20 mg/mL in the 30% ACN [*v*/*v*] containing 0.1% TFA) and 1 µM of oDHF_HW_ were deposited on the MALDI target plate and dried under vacuum.

**Purification of recombinant Aβ42.** The Aβ42 peptide was expressed in *E. coli* BL-21Star^TM^ (DE3) (Invitrogen, Carlsbad, CA, USA) and purified similarly as previously described [21]. In brief, the cell culture was grown for 15 h in a ZYM-5052 medium containing ampicillin (100 µg/mL). Then, after collecting the cell pellet, it was homogenized, sonicated, centrifuged and washed using 20 mM Tris/HCL and 1 mM EDTA buffer solution (pH 8.0). The procedure was repeated three times to remove all soluble proteins. Then, the remaining cell pellet was resuspended, homogenized and sonicated in the buffer solution containing 20 mM Tris/HCL, 8 M urea and 1 mM EDTA (pH 8.0). After centrifugation, the supernatant was diluted four times using 20 mM Tris/HCL, 1 mM EDTA buffer solution (pH 8.0) and mixed with 50 mL DEAE-Sepharose (equilibrated in the same buffer solution), and agitated using a magnetic stirrer at 80 rpm for 30 min at 4 °C. For the chromatography procedure, a Buchner funnel with Fisherbrand glass microfiber paper on a vacuum glass bottle was used. The resin was washed with the loading buffer (20 mM Tris/HCL, 1 mM EDTA pH 8.0), increasing NaCl concentration using a step gradient procedure. The steps contained 0, 20, 150, and 500 mM of NaCl in the loading buffer solution. The target protein was collected in the fraction containing 150 mM NaCl. Collected protein sample was flash-frozen, lyophilized and stored at −20 °C.

To separate any impurities from the target peptide, size exclusion chromatography (SEC) was performed. The lyophilized powder was dissolved in a 20 mM sodium phosphate buffer solution containing 5 M GuSCN (pH 8.0). The sample was then loaded on a Tricorn 10/300 column (packed with Superdex 75) and eluted at 1 mL/min using 20 mM sodium phosphate, 0.2 mM EDTA and 0.02% NaN_3_ buffer solution (pH 8.0). The target peptide was eluted at around 15 min. The eluted protein fraction was lyophilized and stored at −20 °C. Before each experiment, the SEC procedure was repeated to prepare the fresh peptide solution (without any oligomeric or fibrillar structures). The protein fraction with purified Aβ was collected to the maximum recovery microtubes (Corning, New York, NY, USA) and stored on ice for 2 min, while the concentration was determined by integrating the UV absorbance peak (ε_280_ = 1490 M^−1^ cm^−1^). Then, the peptide sample was diluted and immediately used for the experiments.

**Purification of recombinant aSyn.** aSyn was purified similarly as previously described [35]. In short, the cells with expressed aSyn were homogenized and then sonicated using a buffer solution containing 20 mM Tris/HCl, 0.5 M NaCl, 1 mM EDTA and 1 mM PMSF (pH 8.0). The supernatant was then heated for 20 min at 80 °C using a water bath. The aggregated proteins were centrifuged at 15,000× *g* for 30 min at 4 °C, and then soluble aSyn (collected in the supernatant) was precipitated by adding ammonium sulphate (using ~40% saturation). The precipitated proteins were centrifuged and dissolved in a dialysis buffer solution containing 20 mM Tris/HCl, 1 mM EDTA, 0.5 mM DTT, and were dialyzed and loaded onto the HiScale 26/300 column (packed with DEAE-Sepharose equilibrated in the dialysis buffer). The target peptide was washed using a salt gradient (0.1 M NaCl). The collected fractions with aSyn were concentrated and loaded on HiLoad 26/600 column (packed with Superdex 75 pg) and eluted at 4 mL/min using a 50 mM ammonium bicarbonate buffer solution. The eluted protein was flash-frozen, lyophilized and stored at −20 °C.

**Aggregation kinetics of Aβ peptide.** The purified peptide (1.5 mL, pH 8.0) was mixed with 3 mL 20 mM sodium phosphate buffer solution (pH 6.33) to yield a 3-fold diluted peptide solution (pH 7.0). The peptide was mixed with 20 mM sodium phosphate buffer (pH 7.0), 10 mM thioflavin-T (ThT) (Sigma-Aldrich, cat. No. T3516) stock solution, oxidation buffer and oDHF (or its fraction–oDHF_LW_, oDHF_HW_) to the final reaction mixture, containing 1 µM Aβ, 20 µM ThT, 10% (*v*/*v*) (percentages in all further cases are in form of *v*/*v*) of corresponding oDHF sample and 1% DMSO. For the experiments where a range of oDHF_HW_ concentrations were used (0–20%), the initial oDHF_HW_ was diluted using an oxidation buffer. The aggregation kinetics was followed in non-binding 96-well plates (Fisher, Waltham, MA, USA, cat. No. 10438082) (sample volume was 80 µL) at 37 °C by measuring ThT fluorescence using 440 nm excitation and 480 emission wavelengths in a ClarioStar Plus plate reader (BMG Labtech, Ortenberg, Germany).

**Aggregation kinetics of aSyn.** Lyophilised aSyn powder was dissolved in a 20 mM potassium phosphate buffer (pH 7.4) containing 300 mM NaCl and filtered through a 0.22 µm syringe filter. The protein concentration was determined by measuring the sample absorbance at 280 nm using a Nanodrop 2000 spectrophotometer (Thermo Fisher Scientific, Inc., Waltham, MA, USA) (ε_280_ = 5960 M^−1^ cm^−1^). Then, the protein sample was mixed with 10 mM ThT stock solution, oxidation buffer and oDHF (or its fraction–oDHF_LW_, oDHF_HW_) to a final reaction mixture, containing 100 µM aSyn, 100 µM ThT, 20% of corresponding oDHF sample and 2% DMSO. In case of aSyn fibril samples that were used with SH-SY5Y cells, the range of inhibitor concentration was used (0.1–40%). The kinetic experiments were performed similarly as mentioned above with Aβ additionally using 600 RPM orbital agitation and each plate well containing a single 3 mm glass bead (sample volume was 100 µL).

**Atomic force microscopy (AFM).** Samples for AFM images were collected after kinetic measurements (experiment time-1000 min for Aβ and 2500 min for aSyn) and scanned similarly as previously described [20]. In short, 40 µL of 0.5% (*v*/*v*) APTES (Sigma-Aldrich, cat. No. 440140) in MilliQ water was deposited on freshly cleaved mica and incubated for 5 min. Then, the mica was carefully washed using 2 mL of MilliQ water and dried under gentle airflow and 40 µL of each sample was deposited on the functionalized surface and incubated for another 5 min. Prepared samples were again washed using 2 mL of MilliQ water and dried under gentle airflow. AFM imaging was performed using a Dimension Icon (Bruker) atomic force microscope in tapping mode using RTESPA-300 probes. The 10 µm × 10 µm images of the 1024 × 1024 pixel resolution were analyzed using Gwyddion 2.5.5 software. The heights and widths of the structures found on the mica were determined by tracing perpendicular to each structure axes.

**FTIR.** aSyn fibrils were pelleted by centrifuging at 16,100× *g* for 10 min. Then, the supernatant was removed, and 0.5 mL of D_2_O was added. This procedure was repeated three times to replace most of the water molecules with the D_2_O. After the last centrifugation step, the fibrils were resuspended in 80 µL of D_2_O. The FTIR procedure was performed as previously described [36]. In brief, the spectra were recorded using an Invenio S IR spectrophotometer equipped with a Mercury Cadmium Telluride detector. The sample was placed in the CaF_2_ transmission windows with 0.05 mm Teflon spacers, 256 interferograms of 2 cm^−1^ resolution were averaged per spectrum. Before normalizing to the same area of Amide I region (1705–1595 cm^−1^), the D_2_O and water vapor spectrums were subtracted. The data were processed using GRAMS software (Thermo Fisher Scientific, Inc., Waltham, MA, USA).

**Cell culturing.** SH-SY5Y human neuroblastoma cells were obtained from American Type Culture Collection (ATCC, Manassas, VA, USA). The cells were grown in Dulbecco’s Modified Eagle Medium (DMEM) (Gibco, Grand Island, NY, USA), supplemented with 10% Fetal Bovine Serum (FBS) (Sigma-Aldrich, St. Louis, MO, USA), 1% Penicillin-Streptomycin (10,000 U/mL) (Gibco, Grand Island, NY, USA) at 37 °C in a humidified, 5% CO_2_ atmosphere in a CO_2_ incubator.

**MTT assay.** SH-SY5Y cells were seeded in a 96-well plate (15,000 cells/well) and were allowed to attach overnight. Then the medium was changed to the one containing Aβ monomers or fibrils or aSyn fibrils with or without the different amounts of oDHF_HW_ (2% to 20% of the three times concentrated sample) and different amounts of oDHF_HW_ (2% to 20% of the three times concentrated sample) alone. The cell viability was not performed with aSyn monomers due to the aggregation conditions-constant agitation with glass beads which may disrupt the assay. To remove DMSO, the oDHF_HW_ sample was lyophilized and resuspended in a 10 mM phosphate buffer (pH 8.0). Medium, containing buffers (20 mM phosphate buffer solution as Aβ stock solution and 10 mM phosphate buffer as oDHF_HW_ stock solution), served as a control. After 48 h of incubation 10 µM of 3-(4,5-dimethylthiazol-2-yl)-2,5-diphenyltet-tetrazolium bromide (MTT) reagent (12.1 mM in PBS) was added to each well and left to incubate for 2 h. Then, 100 µL of 10% SDS with 0.01 N HCl solution was added to each well to dissolve formazan crystals and, after 2 h, the absorbance at 570 nm and 690 nm (reference wavelength) of each well was measured using a ClarioStar Plus plate reader.

**LDH assay.** Quantification of LDH release into the medium was assessed using a cytotoxicity detection kit (Roche Applied Science, Germany) under the manufacturer’s protocol. Briefly, SH-SY5Y cells were seeded in a 96-well plate (15,000 cells/well) and were allowed to attach overnight. Then the medium inside the wells was aspirated, and 100 µL of Advanced DMEM (Gibco, Grand Island, NY, USA) was added to each well. Afterwards, Aβ monomers with different oDHF_HW_ concentrations were suspended in Advanced DMEM and were added to the wells (to reach 200 µL total medium volume in each well), which resulted in the final 3 µM Aβ and range of oDHF_HW_ (2–20%) concentrations. After 24 h of incubation, 100 µL of the medium from each well was aspirated, centrifugated, and transferred into a TPP 96-well tissue culture test plate (Trasadingen, Switzerland). LDH reagent was added, and after 30 min of incubation at room temperature, the absorbance of wells was measured at 492 nm and 600 nm (reference wavelength) using a ClarioStar Plus plate reader.

**Data analysis.** The aggregation kinetics were followed at least in triplicate independent samples. The data analysis was performed by fitting the kinetic curves using Boltzmann’s sigmoidal equation [36]. The relative halftime values were calculated based on the control sample in their specific microplate. The data were processed using Origin software (OriginLab, Northampton, MA, USA).

All the experiments with SH-SY5Y cells were performed at least in triplicate independent measurements. The obtained values are represented as mean with standard deviation. Student’s t-test was used to evaluate statistical significance between the groups with a probability of * *p* < 0.05, ** *p* < 0.01, and *** *p* < 0.001.

## 3. Results

It is known that the autoxidation of polyphenolic compounds may lead to the formation of the higher mass molecules as a result of oxidation and interflavic coupling reactions [31]. Due to this, we first separated coupling and degradation products of the oxidized 2′,3′-dihydroxyflavone (oDHF) mixture using a concentrator (3 kDa cut-off). The 2′,3′-dihydroxyflavone, oDHF and its fractions of lower molecular weight (oDHF_LW_) and higher molecular weight (oDHF_HW_) components were analyzed using HPLC separation (Figure 1A). The 2′,3′-dihydroxyflavone autoxidized to many different components seen in the chromatogram. The shallow and broad peak (12–27 min) resembles the elution of the mixture containing various lengths of oligomeric molecules [37]. Most of the eluting peaks in the oDHF_LW_ sample separation were within the 2.5–17.5 min range. In the case of oDHF_HW_, the chromatogram mainly contains a shallow peak that is typical for a broad range of oligomeric species (12–27 min). Except for this, the chromatogram contains a few peaks that are also visible in the oDHF_LW_ chromatogram.

To further test the inhibitory effect of each sample, the oDHF, oDHF_LW_ and oDHF_HW_ were used in the aSyn (Figure 1B,D,F) aggregation experiment. When using oDHF, the aggregation is stochastic (t_50_ values are highly dispersed) and reaches plateau at twice as low ThT intensity values compared to the control sample. Similar diminished ThT intensity values were seen in the experiments with oDHF_HW_, where the relative halftime was the highest amongst the tested inhibitor samples. However, the oDHF_LW_ had the lowest effect on aSyn aggregation, only slowing down the aggregation by ~25%. A similar result is seen in the aggregation experiments with Aβ (Figure 1C,E,G). The oDHF and oDHF_HW_ greatly diminished the ThT intensity, while oDHF_LW_ had no effect. This effect of ThT quenching, which has no impact on or correlation with the number of aggregates formed, was previously assessed in the literature [21]. The relative halftime values (Figure 1H,I) followed the same trend, where the oDHF_HW_ showed a significantly higher inhibitory effect than the oDHF. At the same time, the oDHF_LW_ did not possess any influence on the aggregation time of Aβ.

Based on the results presented in Figure 1, we have further investigated the oDHF_HW_ sample. First, we measured the mass spectrum with MALDI-TOF using positive ionization and two ionization matrices (Sinapinic acid and 2,5-DHB) (Appendix A). The results revealed a number of different *m*/*z* values ranging from 400 *m*/*z* to 2000 *m*/*z*. The m/z difference between maxima of adjacent peak clusters was measured to be constant (~246.51 ± 0.36 molar mass (the Z value was determined to be equal to one)). The visible pattern of *m*/*z* distribution is specific to the polymeric structure. It may be that lower molecular weight compounds did oversaturate the detector, limiting the observation of higher molecular weight polymers. It is worth noting that the lowest *m*/*z* values did not match the mass of 2′,3′-dihydroxyflavone (254,238), nor the mass difference between different lengths of polymers. In addition, the 2′,3′-dihydroxyflavone and oDHF_HW_ NMR spectra display different characteristics (Appendix A). The unoxidized flavone NMR spectrum exhibits intense and narrow peak signals assigned to the initial flavone structure. The opposite spectrum is seen in the case of the oDHF_HW_ sample. A broad multiplet in the aromatic region is observed (between 6.33 and 8.68 ppm), which indicates a formation of potentially branched polydisperse polymers. The high level of polydispersity of oDHF_HW_ was confirmed using DLS measurement data (data not shown). Such findings correlate with eight aromatic protons (atoms 1–3, 6, 9, 13–15) observed in 2′3′-dihydroxyflavone NMR spectra (Appendix A). Protons of hydroxyl groups (18 and 19, 10.00 and 9.57 ppm respectively) were not present in the proton NMR spectra of oDHF_HW_.

To characterize the inhibitory effect of oDHF_HW_ against Aβ aggregation, a range of inhibitor concentrations was used (Figure 2A). The ThT fluorescence intensity was diminished with the increasing concentration of oDHF_HW_. The ThT fluorescence intensity was lowered by half using 1% of oDHF_HW_ and was entirely reduced to the noise level when the oDHF_HW_ concentration was 20% (Figure 2A and Appendix A). However, the Aβ aggregation relative halftime relationship with the oDHF_HW_ concentration (Figure 2D) followed two different trends that had a point of discontinuity at 2% of oDHF_HW_ concentration. The lower concentrations slightly decrease (up to 10%) the aggregation halftime, enhancing the fibrillization rate. The inhibitory effect, which seems to have affected both nucleation and elongation stages, occurred only when using a 4% or higher inhibitor concentration. The 20% oDHF_HW_ relative aggregation halftime value was excluded from the graph because of the low ThT fluorescence intensity that would result in low-quality fit (Appendix A).

The SH-SY5Y cell viability (Figure 2B) and LDH release (Figure 2E) tests were performed to understand whether the inhibition of Aβ aggregation also reduces its cytotoxicity to cells. The oDHF_HW_ alone did not significantly affect the cell viability and LDH release at any used concentration. Despite that, when Aβ monomers and oDHF_HW_ were incubated with cells, the survivability adopted the non-linear correlation. When the oDHF_HW_ concentration was 2%, the cell survivability and LDH release were the same as without the inhibitor. Increasing the concentration to 4% increased the cell viability to a statistically significant (*p* < 0.05). Nonetheless, the higher oDHF_HW_ concentration turned in the opposite direction, where the cell viability was lower than with Aβ alone.

The equal effect is seen when Aβ was aggregated with oDHF_HW_, and then the preformed fibrils were added to the SH-SY5Y cells medium and incubated for 48 h (Figure 2C). At the same time, there was no significant effect on the cell viability if the fibrils were formed without oDHF_HW_ and only then mixed together and incubated (24 h at 37 °Celsius) before pooling them on cells. More favorable results were observed with the preformed fibrils of aSyn in the presence of a different concentration of oDHF_HW_ (Figure 2F). When the oDHF_HW_ concentration was 1% to 2%, a significant reduction (*p* < 0.001) of aSyn cytotoxic effect on SH-SY5Y cells was recorded. The higher concentration of inhibitors had less of an impact, although it significantly increased cell viability.

Atomic force microscopy imaging was employed to observe the structures formed during Aβ aggregation kinetics experiment in the presence of oDHF_HW_ (concentration range 0–20%) (Figure 3A–E). The Aβ alone formed typical short 4–5 nm height fibrils that can be seen in the aggregation experiments using a low peptide concentration (1 µM) [38,39]. Very similar structures were observed with 1% of oDHF_HW_. However, the height (Figure 3G) and width (Figure 3H) distribution are more dispersed since oDHF_HW_ (as a polymeric structure) (Figure 3F) was found on the mica in the form of round shape structures with height (4–15 nm) and width (20–40 nm) values larger than the Aβ alone. In the sample images where the oDHF_HW_ concentration was higher (4% to 20%), the larger round shape structures were visible, potentially being the product of the Aβ and oDHF_HW_ combination. These structures are statistically seen in the upward shift of maximum height and width mean and median values. No round shape structures, nor significant height or width distributions, were found in the samples of aSyn aggregates formed in the presence of oDHF, oDHF_LW_ and oDHF_HW_ (Appendix A).

## 4. Discussion

The interest in the autoxidation of polyphenolic molecules in protein aggregation studies is growing because it has been proven that the initial polyphenol does not have an effect against the aggregation process, while the autoxidation mixture possesses anti-amyloid features [20,21,22,26]. The 2′3′-dihydroxyflavone autoxidation was confirmed in our study (Figure 1A), leading to the scarce amount of the initial compound and a variety of different end-products. These products were possibly formed during the processes of degradation, oxidation coupling and polymerization [30,33,34].

The initial strategy of using membrane concentrators was expected to aid in isolating the larger molecules fraction (considering the formation of polymeric structures) from the rest of the smaller molecules in the autoxidation sample. There is no real evidence about whether all oDHF_HW_ molecules at the time of separation were more than 3 kDa molecular weight. In fact, the smaller molecules (<3 kDa) might not have passed through the membrane (3 kDa cut-off) because of the possible planar aromatic structure that cannot bend through the multiple pores. Despite this fact, the separated fractions exhibited drastically different inhibition effects against protein aggregation (Figure 1B–I). The oDHF_HW_ had sustained all the inhibitory potential, while oDHF_LW_ only slightly impacted aSyn aggregation, providing the evidence that larger flavone autoxidation products can function as anti-aggregation agents. This may also reflect the idea that a different concentration of inhibitor is necessary for a significant inhibition of distinct protein/peptide aggregation processes. In particular, this inhibitor/protein ratio is significantly lower for aSyn compared to studies with Aβ [27]. For this reason, the lower amount of inhibitor that may be present in DHF_LW_, is capable of slowing down the aggregation of aSyn, but not Aβ. In the case of Aβ, the molecular mass of the inhibitor must be larger than what passes through the 3 kDa membrane, while for aSyn, smaller polymeric molecules (that do pass through the membrane) also act as inhibitor compounds. This led to the assumption that only larger molecular mass components of the autoxidation sample possess the anti-amyloid features.

The larger mass molecules from the DHF autoxidation sample adopted polydisperse polymeric characteristics, loss of the hydroxyl groups and coordination with sodium ions (the mass of neighboring peaks differ by 23 Da (Appendix A)). These findings indicate that the autoxidation of 2′,3′-dihydroxyflavone (pH 8.0) leads to the polymerization of intermediate products. The literature describes a potential pathway through the formation of quinone moiety in the B ring [30,33]. Such motifs are more likely to form when flavones possess the C2-C3 double bond and absence of 3-OH group. Flavones that contain the aforementioned characteristics and neighboring hydroxyl groups are also shown to oxidize and exhibit anti-amyloid features [21]. However, the MALDI-TOF results suggest that the polymerization mechanism is more complex. The measured monomeric mass difference (246–247 Da) leads to an assumption that the polymerization occurs after further molecular changes or coupling of degradation products.

Since the inhibitor is a polymer, it is necessary to discuss its impact on protein aggregation. The AFM data in this work suggest the formation of higher and wider round-shaped structures. These macromolecules might be the reaction products between the inhibitor and a certain form of Aβ structure (monomer/oligomer or fibrillar) (Figure 3). The increasing concentration of oDHF_HW_ reduces the aggregation rate; however, this trend does contradict the cell viability assay. This inconsistency between results may be related to ThT fluorescence quenching by the inhibitor molecules, which reduces the signal intensity but not the formation of fibrillar structures. The longer aggregation term may lead to the formation of different Aβ aggregation intermediate products through a changed aggregation pathway that resulted in different cytotoxic effects on the SH-SY5Y cells (Figure 2B,E). In fact, the preformed fibrils with oDHF_HW_ showed similar cell viability results, meaning that the cytotoxic effect is exhibited by the end products. At the same time, the cytotoxicity tends to be gained during the initial protein aggregation phase as oDHF_HW_ did not change the Aβ cytotoxicity to cells when the fibrils were preformed without the inhibitor. On a positive note, the lower inhibitor concentrations reveal the reduced cytotoxicity of Aβ and aSyn. This outcome could probably be related to stabilization of protein monomeric/oligomeric species. Potentially, the 4% concentration of oDHF_HW_ in Aβ aggregation experiments leads to the formation of longer protofibrils that are covered with the inhibitor molecules, while at higher oDHF_HW_ concentrations, there are more inhibitor molecules that stabilize smaller oligomeric particles that are more toxic. Therefore, the margin of inhibitor concentration that diminishes the cytotoxic effect is narrow (at lower concentrations, there is no effect, while at higher there is decreased cell viability).

## 5. Conclusions

Taking all these results together, it appears that only part of the autoxidation products of 2′,3′-dihydroxyflavone exhibit anti-amyloid effects against Aβ and aSyn. These molecules appear to be of a polymeric nature and bind to the oligomeric and fibrillar structures of Aβ increasing the cell viability; however, the effect is reversed at higher concentrations of inhibitor.

## Figures and Tables

**Figure 1 antioxidants-11-01711-f001:**
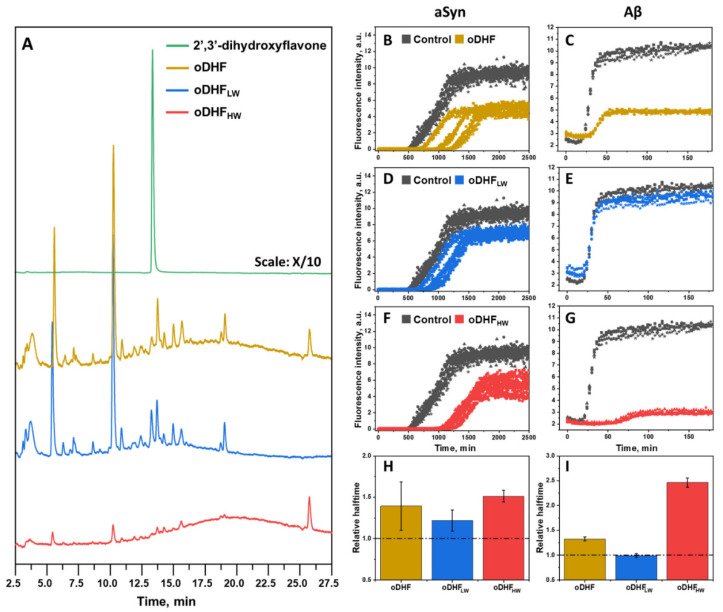
Chromatographic separation of 2′,3′-dihydroxyflavone, oDHF and oDHF fractions (oDHF_LW_ and oDHF_HW_) (**A**). aSyn (100 µM) (**B**,**D**,**F**) and Aβ (1 µM) (**C**,**E**,**G**) aggregation kinetics in the presence of oDHF, oDHF_LW_ and oDHF_HW_ (20% for aSyn and 10% for Aβ experiments) and their respective relative halftime values (**H**,**I**). The relative halftime values were calculated based on the control sample and error bars are one standard deviation (n = 3).

**Figure 2 antioxidants-11-01711-f002:**
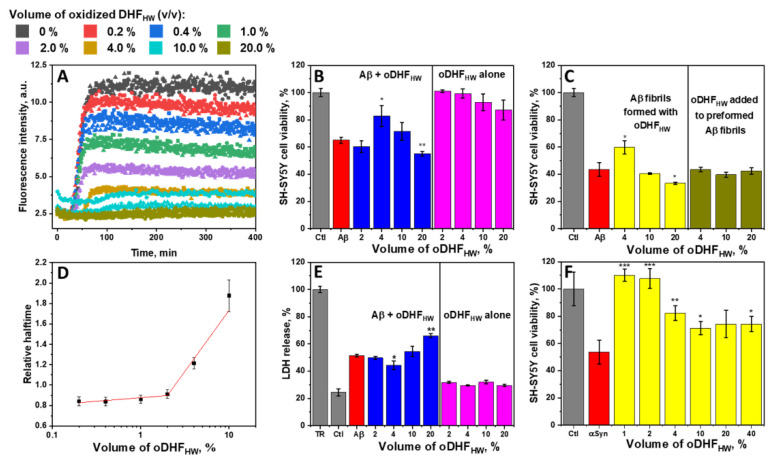
The kinetic curves of 1 µM Aβ aggregation with a range of oDHF_HW_ (0%–20% *v*/*v*) (**A**), and their respective relative halftime values (**D**) compared to the control sample. SH-SY5Y cell viability (**B**) and LDH release (**E**) after incubating (48 and 24 h, respectively) with 3 µM of Aβ and different oDHF_HW_ concentrations (% by *v*/*v*). SH-SY5Y cell viability after incubating 48 h with 3 µM Aβ (**C**) and 100 µM aSyn (**F**) fibrils formed with or without the oDHF_HW_. Student’s t-test significance values were compared to the Aβ or aSyn controls, * *p* < 0.05, ** *p* < 0.01, and *** *p* < 0.001. Relative aggregation halftime error bars are one standard deviation (n = 3) approximated using a linear fit.

**Figure 3 antioxidants-11-01711-f003:**
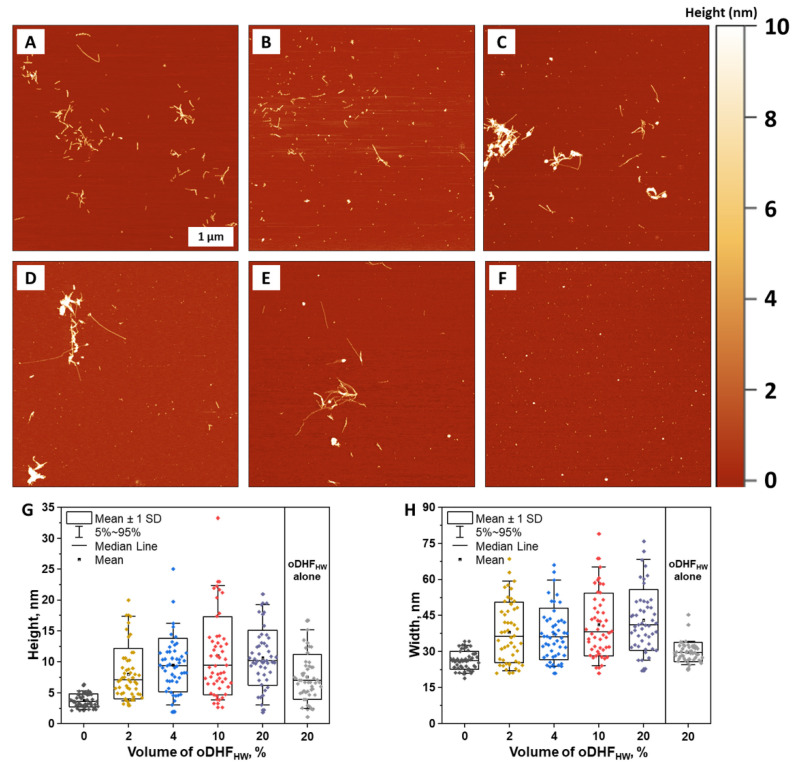
Atomic force microscopy (AFM) images of Aβ without (**A**) and with 1% (**B**), 4% (**C**), 10% (**D**) and 20% (**E**) oDHF_HW_ concentration and the 20% of oDHF_HW_ alone (**F**). The height (**G**) and width (**H**) distribution of structures present on the mica, where box plots indicate mean ± SD and error bars are in the 5–95% range (n = 50).

## Data Availability

The data presented in this study are available on request.

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
