# Peer review of "Exploring the Formation of Polymers with Anti-Amyloid Properties within the 2′3′-Dihydroxyflavone Autoxidation Process"

_antioxidants, 2022, doi:10.3390/antiox11091711_

Round 1

Reviewer 1 Report

Dear Editor,

This is an original well written article about the potential of autoxidated products of 2’,3’-dihydroxyflavone as inhibitors of amyloid-beta and alfa-synuclein aggregation. These two proteins are pathological hallmarks of Alzheimer´s and Parkinson´s disease and their aggregation and deposition is know to contribute to disease progression in both scenarios. In the manuscript, the authors show that autoxidation of 2’,3’-dihydroxyflavone results in products of different molecular weight and they have distinct effect on the aggreagation of abeta and asyn. The introduction and method sections are well-written and provide accurate and detailed information. The results and discussion section should be revised in some sections:

-       Fig 1. Is the relative halftime calculated compared to controls?

-       Fig 2A includes a brown line (what is this?) and 2 blue lines.

-       Fig 2D Relative halftime at 100uM is not shown (the graph stops at 64uM).

-       Fig 2B and 2E, it is not clear how is the LDH release experiment performed. Also on ratios abeta and oDHF? What is the concentration indicated below?

-       Why are the results of alfa-synuclein only shown for fibrils?

-       The results of oDHFHMW on alfasynuclein fibrils toxicity is significant. This doesn´t seem to match the ThT results on Fig1. These discrepancies (if they are) should at least be discussed.

-       Like in Figure S4, the article should include the AFM results of oDHFLMW on Abeta too.

-       In the discussion section the authors go through most of the results but do not discuss the possible explanations considering the differences observed between abeta and alfa-synuclein prevention of aggregation by oDHF.

Reviewer 2 Report

The manuscript by Sakalauskas et al. reported that the high-molecular-weight autooxidation products of 2’,3’-dihydroxyflavone (oDHF) exhibit anti-amyloidogenic activities against amyloid-β (Aβ) and α-synuclein (α-syn). This manuscript provides intriguing evidence to understand the molecular mechanism of oDHF for its anti-amyloidogenic activity and important insight into developing novel and potent inhibitors for Aβ and α-syn aggregation. However, I have some major concerns that need to be addressed before considering the publication of this manuscript.

1. It is indeed intriguing that the high-molecular-weight products of oDHF are the most active component in suppressing the aggregation of Aβ and α-syn. However, I think that the manuscript did not provide sufficient details regarding how the amount of each oxidation product of oDHF was measured and treated to the aggregation reaction mixture. For example, the manuscript indicated in the method section that “50uM of corresponding oDHF sample” was used for the Aβ aggregation assay, yet it did not explain how the concentrations of oDHF_LW and oDHF_HW were measured.

2. I think that the authors should provide more detailed information regarding the molecular characteristics of oDHF_HW. For example, could the authors conduct some gel-permeation chromatography, dynamic light scattering, NMR DOSY, or similar analysis to better appreciate the size distributions of oDHF_HW? In addition, it would be helpful if the authors could provide or at least discuss the expected chemical structure model of the polymeric oDHF.

3. The authors should explain/discuss the inconsistency between the in vitro aggregation assay and the cell viability test of Aβ. For example, in the figure 2, it is clear that Aβ aggregation was almost perfectly prohibited with 100uM oDHF_HW, while the same amount of oDHF_HW cannot restore cell viability. I think that the related statement in conclusion (the lines 408~410) is too speculative without further experimental support (such as the binding study between oDHF_HW and Aβ monomers/oligomers). 

Round 2

Reviewer 2 Report

Although the revised manuscript by Sakalauskas et al. addressed some of my previous concerns, I am afraid that the following issues are not fully resolved yet.

1. The authors mentioned in their response that the concentration of the oDHF fractions could not be measured accurately. Then, would it be possible that the higher inhibitory effects of oDHF_HW in the aggregation assay were simply because more oDHF was added to this aggregation mixture? The authors should provide at least some reasonable explanation why they think that oDHF_HW is the most effective inhibitor without accurate control of the added amount.

2. As pointed out by the authors, I think that the possible quenching effects of oDHF fractions may compromise the reliability of all the ThT aggregation assays (Fig. 2). Could the authors complement at least some of these with an additional assay, e.g., turbidity measurement (this could be helpful for the authors to explain their cell viability assay)?

Round 3

Reviewer 2 Report

The authors addressed all of my concerns.